# Effect of long-chain polyunsaturated fatty acids in infant formula on long-term cognitive function in childhood: A systematic review and meta-analysis of randomised controlled trials

**Maximiliane L. Verfuerden**[1]*, **Sarah Dib**[1], **John Jerrim**[2], **Mary Fewtrell**[1], **Ruth E. Gilbert**[1,3]

1 University College London Great Ormond Street Institute of Child Health, London, United Kingdom,
2 University College London Institute of Education, London, United Kingdom, 3 Health Data Research UK, London, United Kingdom

* m.verfuerden@ucl.ac.uk

**Data Availability Statement:** All relevant data are within the manuscript and its Supporting information files.

## Abstract

Lack of preformed long-chain polyunsaturated fatty acids (LCPUFA) in infant formula has been hypothesised as contributing to cognitive differences between breast-fed and formula-fed infants. Previous systematic reviews found no cognitive differences between infants fed formula with LCPUFA and those fed formula without, but focused on early developmental measures, such as Bayley Scales of Infant Development, which are poorly differentiating and not predictive of cognitive ability in childhood. This systematic review examined the effect of randomising infants to formula supplemented with LCUFA vs unsupplemented formula on cognitive function ≥ age 2.5 years. We searched Medline, Embase the Cochrane Central Register of Controlled Trials without date limit, following a pre-published protocol according to PRISMA guidelines. We conducted random effects meta-analyses in RevMan v5.4 and followed GRADE and Cochrane Guidelines to evaluate strength of evidence and potential for bias. We included 8 trial cohorts which randomised participants between 1993 and 2004 and analyse 6 previously unpublished outcomes provided by various trialists. Age at the last available cognitive test ranged from 3.3 to 16 years. The pooled mean difference in Wechsler Preschool and Primary Scale of Intelligence-Revised from four trials in term-born children showed no benefit of LCPUFA: -0.04 points (95% confidence interval -5.94 to 5.85, 95% prediction interval -14.17 to 14.25). The pooled mean difference in Wechsler Abbreviated Scale of Intelligence score from two trials in preterm-born children also showed no benefit of LCPUFA: -7.71 (95% CI -24.63 to 9.22, 95% PI -97.80 to 82.38). Overall quality of evidence was low, due to substantial heterogeneity, low rates of follow-up, and indications of selective publication. The long-term effect of LCPUFA supplementation in term and pre-term-born infants on cognition is highly uncertain and includes potential for large benefit as well as large harm. Based on our findings, LCPUFA supplementation of infant formula is not recommended until further robust evidence excludes long-term harm.

**Funding:** This study was supported by funds from the Economic and Social Research Council, and the Great Ormond Street Hospital Charity. RG was supported by Health Data Research UK, an initiative funded by the UK Research and Innovation, Department of Health and Social Care (England) and the devolved administrations, and leading medical research charities. Research at UCL Great Ormond Street Institute of Child Health is supported by the NIHR Great Ormond Street Hospital Biomedical Research Centre. The funders had no role in study design, data collection and analysis, decision to publish, or preparation of the manuscript.

**Competing interests:** MF has been a member of the Infant Nutrition Working Group at EFSA (European Food Safety Authority) since 2013. She was involved in data analysis and publication of randomised trials of LCPUFA-supplemented infant formulas funded by grants from Numico Res BV and Heinz UK. The companies also provided the infant formulas for the studies. She was also involved in follow-up studies (including cognitive outcome) of children and adolescents from randomised trials of LCPUFA-supplemented formulas, with funding from the Medical Research Council and European Union (FP6-FOOD-2005-007036). MLV, SD, JJ and REG have no competing interests to declare.

**Abbreviations:** AA, arachidonic acid; BBCS-R, Basic Concept Scale-Revised; CI, Confidence Interval; DHA, docosahexaenoic acid; EU, European Union; IQ, intelligence quotient; MD, mean difference; PI, Prediction Interval; PPVT, Peabody Picture Vocabulary Test; RCT, randomised controlled trial; SB IQ, Stanford-Binet IQ; SE, standard error; SMD, standardised mean difference; WASI, Wechsler Abbreviated Scale of Intelligence; WASI, Wechsler Adult Intelligence Scale; WPPSI-R, Wechsler Preschool and Primary Scales of Intelligence-Revised; LCPUFA, long-chain polyunsaturated fatty acid.

## Study registration

PROSPERO registration numbers CRD42018105196 and CRD42018088868.

## Introduction

Long-chain polyunsaturated fatty acids (LCPUFA), such as docosahexaenoic acid (DHA) and arachidonic acid (AA), are important structural components of the human brain that mainly accumulate during the third trimester of pregnancy and early infancy [1–3]. Human breast milk contains DHA, AA, and their fatty acid precursors [4] but, historically, infant formula contained only the precursors alpha linoleic acid and linoleic acid, which infants, especially those born preterm, may not be able to effectively synthesise into DHA and AA [5].

Research suggests that breast-fed children have higher cognitive ability compared to formula-fed children [6–9]. Lack of preformed LCPUFA in infant formula has been hypothesised as contributing to these cognitive differences. Yet so far there is no clear evidence from published randomised controlled trials (RCTs) that LCPUFA-supplemented infant formula improves cognition compared with unsupplemented formula milk [5, 10]. Previous systematic reviews of RCTs may have failed to detect a difference in cognition because they mainly focused on early measures of cognition, such as Bayley Scales of Infant Development. Early measures of cognition are, however, not adequate to differentiate between cognitive skills potentially affected by nutrient supplementation and are poorly predictive of cognition during school age [11–13]. Follow-up later in childhood, using more reliable measures of cognitive function such as Intelligence Quotient (IQ) scores, might be more likely to detect an existing effect of LCPUFA-supplementation.

Clear evidence on the long-term effects of LCPUFA-supplementation is needed as the EU Commission (EC) recently mandated the addition of one type of LCPUFA, DHA, to all infant and follow-on formulae [14]. While the decision acknowledged the lack of evidence on cognitive benefits and was instead based around theoretical arguments, supplementation comes at a cost: a family can spend up to $400 extra per year on LCPUFA-supplemented- compared to unsupplemented infant formulas and mandatory supplementation may result in price rises across the market [15].

To our knowledge, no systematic review has previously focused on later childhood -when more accurate measures are available [11–13]- to study the cognitive effects of infant formula supplemented with LCPUFA. The present study combines published and previously unpublished trial data, acquired through contacting trial authors, to compare the long-term cognitive effects of LCPUFA-supplemented versus unsupplemented infant formula in children born at term and preterm.

## Methods

### Search strategy

This systematic review and meta-analysis follows two published protocols (one for terms and one for preterms) [16, 17], based on the Preferred Reporting Items for Systematic Reviews and Meta-Analyses (PRISMA) guidelines [18]. We searched Medline, Embase, proceedings from major scientific meetings of child nutrition (S2 File) and the Cochrane Central Register of Controlled Trials in October 2019, without date or language restrictions. We reviewed the reference lists of the included studies and traced subsequent publications. We first identified

RCT participant cohorts based on any infant formula supplementation with LCPUFA, independent of whether cognitive outcomes were reported. We then contacted a total of 18 trialists, ethics committees or industry representatives to identify potential unpublished data, clarify study details and to ask whether they knew of any other eligible trials that had measured cognitive outcomes ≥2.5 years of age (**S4 and S5 Tables in** S1 File). The cut-off of ≥2.5 years was based on the age where early development of the prefrontal cortex, the brain region associated with higher cognitive functions is completed [19]. Abstract review was done in duplicate by MV and SD; consensus was achieved by discussion. All extracted data is available within **S1 Table in** S1 File.

## Inclusion and exclusion criteria

We based our selection on trial cohorts rather than publications. We included trial cohorts where infants were given either infant formula supplemented with LCPUFA (DHA alone or DHA together with AA, at any dose) compared with unsupplemented formula. Trial cohorts were eligible if commencement of the intervention began within 2 weeks of birth and they measured cognitive function age ≥ 2.5 years using validated measures including Wechsler and Stanford-Binet IQ scores. We excluded trial cohorts for which we could not find any cognitive outcomes ≥2.5 years of age (irrespective of whether this outcome was published).

## Outcomes and data analysis

The primary outcome was the pooled difference of cognitive ability between supplemented and unsupplemented groups. We decided to use the cognitive test reported most frequently among the included studies rather than a combination of different tests, to increase the interpretability of the primary outcome and decrease heterogeneity. We aimed to use the mean difference (MD) when the cognitive measure was already standardised (e.g. IQ score) so as to increase interpretability, otherwise we would use the standardised mean difference (SMD). We performed separate analyses for term and preterm-born participants because healthy term infants are able to synthesise LCPUFA from fatty acid pre-cursors, whereas preterm babies are born with fewer LCPUFA reserves accumulated in utero and are less able to synthesise LCPUFA than term-born babies. We therefore hypothesised that term/preterm status could modify the effect of supplemented LCPUFA.

The secondary outcome was the pooled SMD of all available cognitive test scores. To not include the same participants multiple times, we only included one score per trial cohort. For this, we used the score at the oldest age available under the assumption that scores at a later age are a more accurate reflection of cognitive ability than scores at an earlier age.

All analyses were performed in RevMan v5.4 and included participants with the relevant outcome in the groups to which they were randomised. We defined statistically significant differences based on a p-value <0.05 and report all summary measures along with 95% confidence intervals and a measure of heterogeneity ($I^2$). We also calculated the prediction interval (PI) to accurately reflect any uncertainty about clinical harms and benefits [20].

## Strength of evidence and risk of bias assessment

We assessed the strength of evidence and risk of bias for each study using GRADE (**S3 Table in** S1 File) and the Cochrane Risk of Bias Tool II, which was also used as the template for data extraction (**S2 Table in** S1 File). Post-hoc, we explored potential for publication bias by plotting the SMDs for all available scores against their standard error (SE), to visualise the relative distribution of published and unpublished outcomes (**S1 Fig in** S1 File).

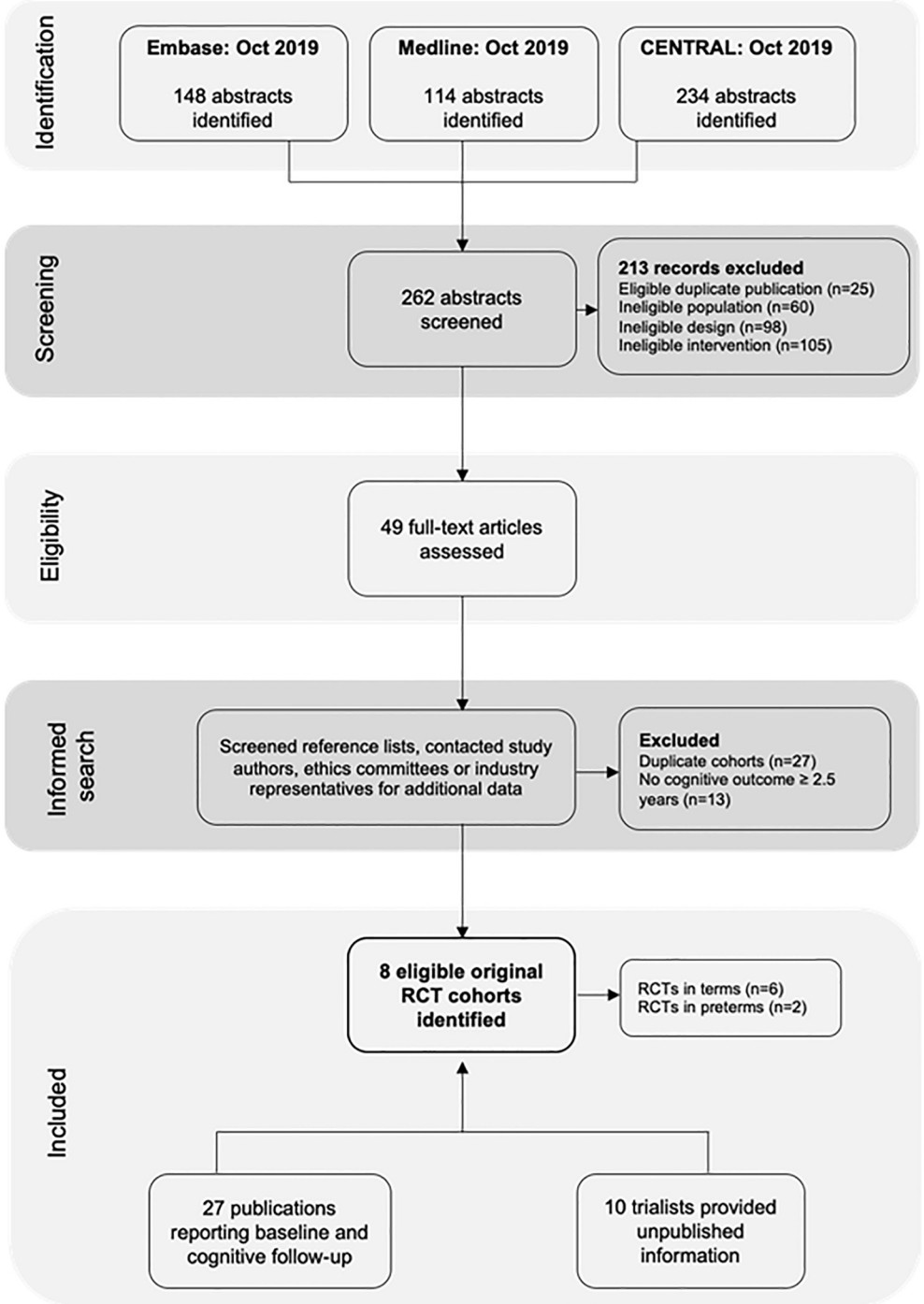

**Fig 1. Study selection process.**

## Results

We included eight unique trial cohorts [21–49] of which six were performed in infants born at term and two in infants born preterm (Fig 1). We obtained previously unpublished outcome data for two RCTs: Firstly, a two-centre trial of term babies in England [30, 31] provided

**Table 1. Characteristics of included RCT cohorts and associated publications.**

| RCT Cohort (recruitment years) and [publication references] | | Criteria | LCPUFA, % | Any breastmilk during study | Breastfed reference group | Start LCPUFA (age in days) | Duration LCPUFA (months) | Outcomes (age in years) | Unpublished outcome data? | %Follow-up: n followed-up[1,2]/ n randomised[2] |
|---|---|---|---|---|---|---|---|---|---|---|
| Term studies | | wga | | | | | | | | |
| US: 3 centres (92–93) [21–23] | | >36 | Egg DHA 0.12 + Egg AA 0.43 | No | Yes | <7 d | 12 | PPVT (3.3) SB IQ (3.3) | | 79% 72/91 |
| Europe: 6 centres* (92–93) [24–29] | | 37–42 | Egg DHA 0.30§+ Egg AA 0.44§ | No | Yes | <3 d | 4 | WPPSI-R (6) | | n/a 147/ n/a** |
| ENG: 2 centres (93–95) [30, 31] | | >36 | Egg DHA 0.32 + Egg AA 0.30 | No | Yes | <7 d | 6 | WPPSI-R (4.5) WASI (16) | Yes | 60% 184/ 309 |
| US: Dallas (93–95) [32–34] | | 37–40 | Algae DHA 0.36 + Fungi AA 0.72 | No | Yes | <5 d | 3.9 | WPPSI-R (4) | | 68% 36/53 |
| NL: Groningen (97–99) [35–38] | | 37–42 | Egg & Fish DHA 0.30 + Fungi AA 0.45 | No | Yes | <5 d | 2 | WASI (9) | partly‡ | 68% 214/ 314 |
| US: DIAMOND [39–46] | Dallas (02–04) | 37–42 | Algae DHA 0.32 + Fungi AA 0.64 | No | No | <10 d | 4 | PPVT (3.5), BBCS-R (2.5) | | 46% 42/92 |
| | Kansas (02–04) | | | No | | | | WPPSI-R (6) | partly‡ | 38% 30/80 |
| Preterm studies | | bw, wga | | | | | | | | |
| ENG: 2 centres (93–96) [47] | | <1750g, <37 | Algae DHA 0.32 + Fungi AA 0.64 | No | Yes | <11 d | 0.69 | WASI (16) | yes | 9% 17/ 196 |
| SCT: Glasgow (95–97) [48, 49] | | ≤2000g, <35 | Egg DHA 0.17 + Egg AA 0.31 | Yes | No | 2–60 d | 9 | WASI (10) | partly‡ | 45% 107/ 238 |

bw: birthweight, wga: Weeks gestational age, PPVT: Peabody Picture Vocabulary Test, SB IQ Stanford-Binet IQ, BBCS-R: Bracken Basic Concept Scale-Revised, WASI: Wechsler Abbreviated Scale of Intelligence, WPPSI-R: Wechsler Preschool and Primary Scale of Intelligence-Revised;

*Two locations unknown, 1: When latest cognitive outcome was measured 2: Only dose/source of interest, some studies had more than one randomised dose/source group (only one per trial is included here);

§Two different concentrations were published: DHA = 0.30 [29] or 0.21 [54]–AA = 0.44 [29] or 0.35 [54])

‡data was published in graphical form or differently modelled; n/a: Not available;

** the 4 centres that were followed up randomised 237 infants between them but it is unknown how many were randomised in the remaining two centres and why they weren't followed-up. It follows that the follow-up rate was < 62%.

unpublished follow-up data on IQ assessments using the Wechsler Preschool and Primary Scales of Intelligence (WPPSI) at age 4.5 years and the Wechsler Abbreviated Scales of Intelligence (WASI) at age 16 years. Secondly, a two-centre trial in England of babies born preterm provided unpublished data from their IQ assessments using the WASI at age 16 years [50]. We also received partly published outcome data from three trials, in a form that allowed them to be included in a pooled meta-analysis: IQ using the WASI at 9 years from a Dutch trial of term babies [35–38]; IQ using the WASI at age 16 years from the Kansas centre of the US based DIAMOND trial conducted in term infants [39, 41–46, 51]; and IQ assessments adjusted for maternal education at 10 years from a two-centre study in Scottish preterm children [48, 49].

Table 1 shows the characteristics of all included trial cohorts. The total number of children randomised was not reported for one RCT [24–29]. Study randomisation was performed between 1992 and 2004 with the latest cognitive assessments conducted at mean ages 3.3–16 years. All studies randomised participants to infant formula supplemented with LCPUFA containing DHA and AA or to unsupplemented infant formula. DHA was sourced from egg, fish, fungi, algae or starflower oil and made up between 0.12 and 0.96% of total fat content. Ratios of DHA:AA ranged from 1:0.8 to 1:3.6. Duration of the intervention ranged from two to 12 months in term infants and three weeks to 9 months in infants born preterm. All trials, except

for one, included a non-randomised breast-fed reference group but these were not analysed in this review. Randomised children in one preterm trial [48, 49] could receive some breastmilk during the first months, but the intake was balanced across groups.

Three RCTs had more than one randomised intervention group [21, 22, 32–34, 39–46, 52]. To ensure comparability with a previous Cochrane review [53], we included only the intervention group in our analysis that was most similar in DHA dose and source to the other included RCTs. One study [39–46] randomised babies in two centres and then conducted different cognitive assessments at different ages for children followed-up stratified by centre. We regarded these as independent and included both reported cognitive assessments in our analysis.

The most frequently reported measures of cognitive function were the WPPSI IQ at ages 4–6 years (in four term RCTs) and the WASI IQ at ages 9–16 years (two term and two preterm RCTs). Among term born participants, assessments at the oldest age comprised WASI at ages 9 and 16 years (2 RCTs), Stanford-Binet IQ at age 3.3 years (1 RCT), the Peabody Picture Vocabulary Test at age 3.5 years (1 RCT) and WPPSI 4–6 years (3 RCTs; see Table 1). Other reported measures in term were the Bracken Basic Concept Scale-Revised at age 2.5 years (one term RCT). Among preterm-born participants, the most frequently reported assessments were also those conducted at the oldest age.

## Primary outcome

We pooled the data for infants born at term using random-effects because heterogeneity was judged to be high with an $I^2$ of 72% (p = 0.01), despite homogeneity in terms of the assessment used. Fig 2 shows that, among term-born babies, the pooled MD from four trials suggests no difference at the 5% level in the WPPSI score between LCPUFA supplemented and control groups: MD -0.04 IQ points. Uncertainty around the effect estimate was extremely high: 95% CI -5.94 to 5.85 and 95% PI from -14.17 to 14.25.

We pooled the data for infants born preterm also using random-effects because heterogeneity was judged to be high with an $I^2$ of 83% (p = 0.01), despite homogeneity in terms of the assessment used. Fig 3 shows that among preterm-born babies, the pooled MD suggests no difference in the WASI Scale IQ at age 9–16 years: MD -7.71 IQ points. Again, uncertainty

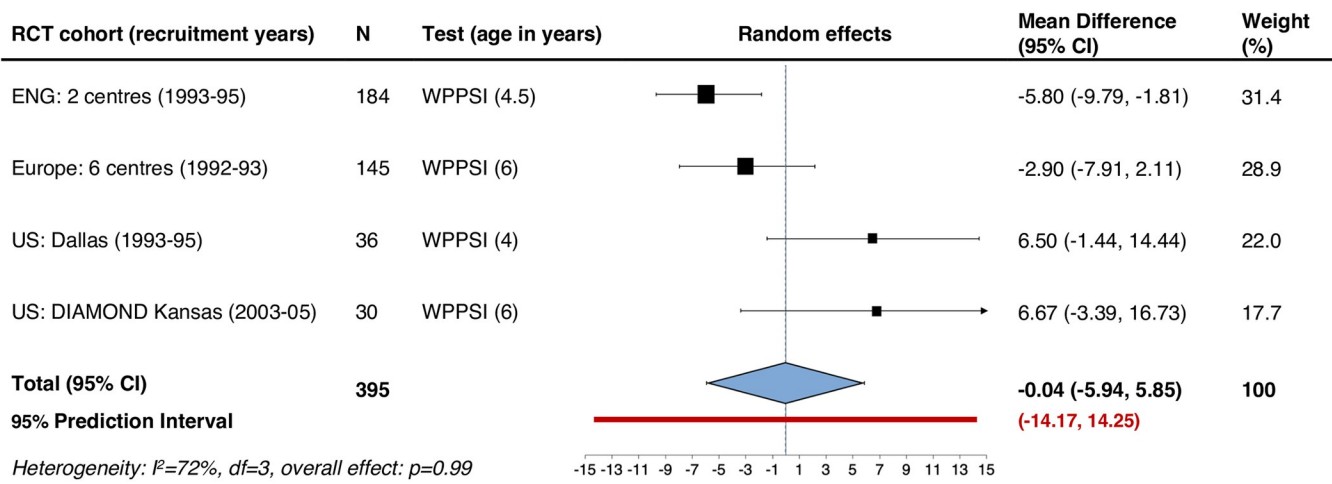

| RCT cohort (recruitment years) | N | Test (age in years) | Random effects | Mean Difference (95% CI) | Weight (%) |
|---|---|---|---|---|---|
| ENG: 2 centres (1993-95) | 184 | WPPSI (4.5) | | -5.80 (-9.79, -1.81) | 31.4 |
| Europe: 6 centres (1992-93) | 145 | WPPSI (6) | | -2.90 (-7.91, 2.11) | 28.9 |
| US: Dallas (1993-95) | 36 | WPPSI (4) | | 6.50 (-1.44, 14.44) | 22.0 |
| US: DIAMOND Kansas (2003-05) | 30 | WPPSI (6) | | 6.67 (-3.39, 16.73) | 17.7 |
| Total (95% CI) | 395 | | | -0.04 (-5.94, 5.85) | 100 |
| 95% Prediction Interval | | | | (-14.17, 14.25) | |

Heterogeneity: *I²=72%, df=3, overall effect: p=0.99*

-15 -13 -11 -9 -7 -5 -3 -1  1  3  5  7  9  11 13 15

Favours unsupplemented    Favours formula + LCPUFA

**Fig 2. Primary outcome term infants: WPPSI-R IQ at ages 4–6 years (mean difference).**

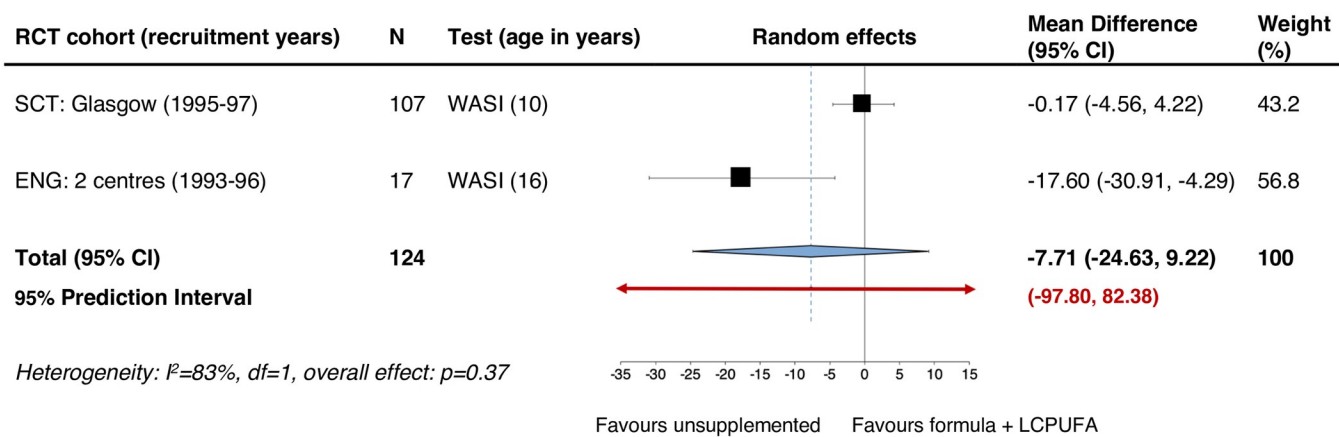

**Fig 3. Primary outcome preterm infants: WASI IQ at ages 10–17 years (mean difference).**

around the effect estimate was extremely high: 95% CI -24.63 to 9.22 and 95% PI from -97.80 to 82.38.

## Secondary outcomes

Pooled cognitive tests scores performed at the oldest available age in each trial showed no evidence that children who received LCPUFA-supplemented infant formula differed from the control group: the SMD for term born children using random effects was -0.10, with a 95% CI of -0.32 to 0.12, with a 95% PI of -0.61 to 0.39 (Fig 4). The cognitive measures included were the Peabody Picture Vocabulary Test (PPVT) at age 3.5 years, Stanford-Binet IQ at 3.3 years, WASI at 9 and 16 years, and the WPPSI at age 4 and 6 years. There were no further available cognitive measures for infants born preterm.

## Strength of evidence and risk of bias assessment

Fig 5 displays the results from the Cochrane Risk of Bias Tool. It highlights that potential for bias from attrition of study participants was a universal problem for all included trials. This is also reflected in the GRADE Summary of Findings (**S3 Table in** S1 File). Overall, the quality of available evidence was low, rated down for heterogeneity, attrition and potential for bias from selective publication. Potential for selective publication was based on correspondence from trialists about the (perceived) difficulty of publishing harmful results. **S1 Fig in** S1 File plots published and unpublished effect estimates against their standard error and indicates that unpublished effect estimates tend to be those that show harm. However, this should be interpreted with caution since visual analysis of potential for publication bias has limited reliability with <10 studies [55]. Completeness of follow-up for cognitive assessment was low, ranging from 9% to 79% of children initially randomised (median 52.6). Most studies reported balanced group characteristics at follow-up (**S2 Table in** S1 File).

## Discussion

We found no evidence that LCPUFA-supplemented infant formula improved long-term cognition among children born at term or preterm. Effect estimates were highly uncertain and included potential for large benefit and large harm.

This uncertainty should be taken seriously. While previous trials on LCPUFA-supplementation mostly reported either no effect or transiently favourable effects on developmental

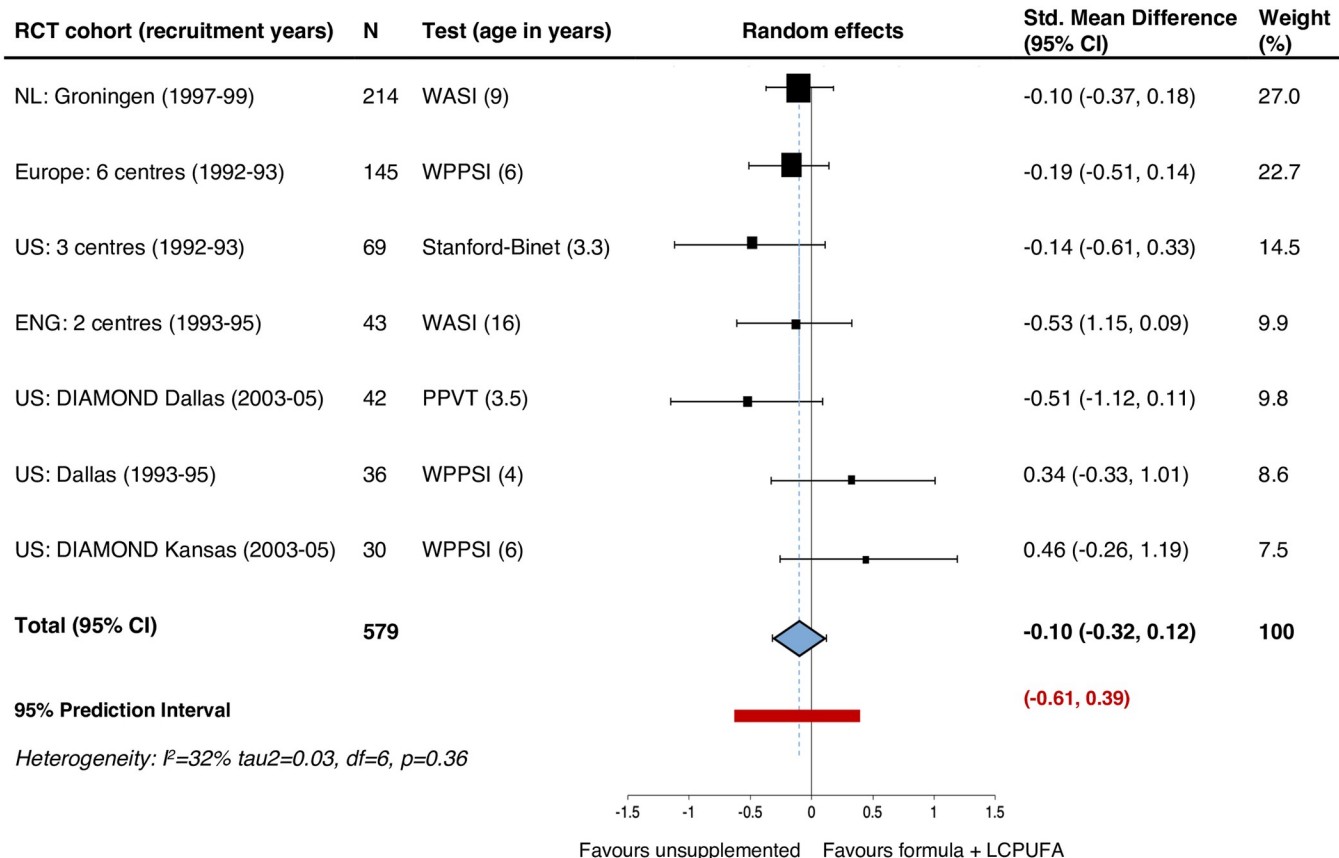

| RCT cohort (recruitment years) | N | Test (age in years) | Random effects | Std. Mean Difference (95% CI) | Weight (%) |
|---|---|---|---|---|---|
| NL: Groningen (1997-99) | 214 | WASI (9) | | -0.10 (-0.37, 0.18) | 27.0 |
| Europe: 6 centres (1992-93) | 145 | WPPSI (6) | | -0.19 (-0.51, 0.14) | 22.7 |
| US: 3 centres (1992-93) | 69 | Stanford-Binet (3.3) | | -0.14 (-0.61, 0.33) | 14.5 |
| ENG: 2 centres (1993-95) | 43 | WASI (16) | | -0.53 (1.15, 0.09) | 9.9 |
| US: DIAMOND Dallas (2003-05) | 42 | PPVT (3.5) | | -0.51 (-1.12, 0.11) | 9.8 |
| US: Dallas (1993-95) | 36 | WPPSI (4) | | 0.34 (-0.33, 1.01) | 8.6 |
| US: DIAMOND Kansas (2003-05) | 30 | WPPSI (6) | | 0.46 (-0.26, 1.19) | 7.5 |
| **Total (95% CI)** | **579** | | | **-0.10 (-0.32, 0.12)** | **100** |
| 95% Prediction Interval | | | | (-0.61, 0.39) | |

Heterogeneity: I²=32% tau2=0.03, df=6, p=0.36

-1.5   -1   -0.5   0   0.5   1   1.5

Favours unsupplemented    Favours formula + LCPUFA

**Fig 4. Secondary outcome term infants: Cognitive function summary score (standardised mean difference).**

outcomes, negative effects are not without precedent. LCPUFA-supplementation has been associated with adverse effects on growth (including head growth) [56–58] and on development such as reduced vocabulary scores in term infants at age 14 months [52]. Furthermore, LCPUFA formula supplementation comprising DHA has been associated with potential harms in other domains, for example a higher risk of bronchopulmonary dysplasia in preterm infants [59].

Potential harms of LCPUFA might relate to the source of LCPUFA. LCPUFA in the included studies of this review were derived from egg, fish, algae and fungi and may not have the same functional effects as LCPUFA present in human milk. Furthermore, the DHA content of human milk is variable and heavily influenced by maternal diet. It therefore does not easily translate into an optimal dose [60]. This incertitude was reflected in the variety of doses administered in the included trials and likely drove the substantial heterogeneity that was observed in our meta-analyses. The plotted SMDs of all available outcomes also suggest that heterogeneity between studies is partly due to under-representation of outcomes showing a harmful effect of LCPUFA. We cannot determine the reason for non-publication for all trials, but a (perceived) difficulty in publishing negative results might play a role. Apart from that, trials were conducted in similar populations, using similar inclusion criteria, and the pooled primary outcomes were based on the same respective test.

We were not able to perform subgroup analyses to determine which factors accounted for the observed heterogeneity due to the small number of available trials. Yet even if findings are

| RCT | Randomisation process | Deviations from intended interventions | Missing outcome data (attrition bias) | Measurement of the outcome | Selective reporting (information bias) |
|---|---|---|---|---|---|
| Term: US, 3 centres (1992-93) | − | ? | − | ? | ++ |
| Term: Europe, 6 centres (1992-93) | − | ++ | − | ++ | − |
| Term: ENG, 2 centres (1993-95) | ++ | ? | − | ++ | ++ |
| Term: US, Dallas (1993-95) | ++ | ++ | − | ++ | ++ |
| Term: NL, Groningen (1997-99) | ++ | ++ | − | ++ | ++ |
| Term: US, DIAMOND Dallas (2002-04) | ++ | ++ | − | ? | − |
| Term: US, DIAMOND Kansas (2002-04) | ++ | ++ | − | ++ | ++ |
| Preterm: ENG, 2 centres (1993-96) | ++ | ++ | − | ++ | ++ |
| Preterm: SCT, Glasgow (1995-97) | ++ | ++ | − | ++ | ++ |

**Fig 5. Risk of bias summary: Based on latest available included cognitive outcome.**

truly inconsistent, this would not change the conclusion of this study, namely that current evidence for supplementing infant formula with LCPUFA on the basis of cognitive benefits is weak and does not exclude potential for large harm.

Strengths of this review include a comprehensive search independent of reported outcomes, duplicate assessment of eligibility, and rigorous application of the GRADE and Cochrane Risk of Bias approach to rate quality of evidence and risk of bias. In contrast with previous systematic reviews, we included previously unpublished outcomes, which enabled a meta-analysis of measures of cognitive function beyond the first two years of life. As the child ages, cognitive assessments become more discriminatory and predictive of adult function than tests before two years of age and less dependent on the situation and outcome assessor [11–13].

The limitations of our review are related to the quality of the underlying evidence. We observed substantial statistical heterogeneity, potential for attrition bias in all outcomes, as well as potential for selective publication. Attrition of study participants is a universal problem in long-term nutrition studies [56–58]. Potentially negative effects of LCPUFA-supplementation could have resulted from selective follow-up. However, it seems unlikely that either 1) children with lower IQ who were previously assigned to the control group would be less likely to respond than children with lower IQ from the intervention group, or 2) children with

higher IQ outcomes be more likely to respond if they were in the control group, especially as all trials were blinded.

While there was industry involvement in all of the original trials, only four of the ten follow-up studies reported in our analyses received industry funding. In six of the eight trials there were other potential conflicts of interest (**S7 Table in** S1 File). While industry involvement is very common in the area of infant nutrition studies it is not necessarily predictive of lower study quality. It is also necessary to emphasise that many of the outcomes included in our review were published at a time when reporting standards were lower than today. Importantly, the included studies represent the only available evidence on long-term cognitive outcomes.

More robust evidence of benefit, and certainty about absence of harm, is needed to justify mandatory LCPUFA-supplementation of infant formula. New trials would take time, are expensive and would suffer from the same problems of attrition and resulting biases as the studies included in this review. Recent methods, involving data linkage of extant trial data to administrative education and health data in adolescence and adulthood, offer a more rapid, less biased, and cost effective way of obtaining data on long-term outcomes. Linkage of historical trials to administrative education or health datasets is achievable where trial data and participant identifiers have been retained and governance arrangements allow secure linkage without re-consent [61, 62].

We found no evidence that LCPUFA-supplemented infant formula benefits cognitive function compared with unsupplemented formula in children born at term or preterm. Given the lack of benefit on other functional outcomes [5, 53] and the additional costs of supplemented formula [63], widespread addition of LCPUFA to infant and follow-on formula cannot be supported until further robust evidence excludes potential for future harm.

## Supporting information

**S1 File.**
(DOCX)

**S2 File.**
(DOC)

## Acknowledgments

The authors thank the investigators Nancy Auestad, Bridget Barrett-Reis, Eileen Birch, Susan Carlson, John Colombo, Mijna Hadders-Algra, Alan Lucas and Atul Singhal for contributing data or providing additional information.

## Author Contributions

**Conceptualization:** Maximiliane L. Verfuerden.

**Data curation:** Maximiliane L. Verfuerden.

**Formal analysis:** Maximiliane L. Verfuerden.

**Funding acquisition:** Maximiliane L. Verfuerden, Ruth E. Gilbert.

**Investigation:** Maximiliane L. Verfuerden.

**Methodology:** Maximiliane L. Verfuerden, John Jerrim.

**Project administration:** Maximiliane L. Verfuerden.

**Resources:** Maximiliane L. Verfuerden.

**Supervision:** John Jerrim, Mary Fewtrell, Ruth E. Gilbert.

**Validation:** Sarah Dib.

**Visualization:** Maximiliane L. Verfuerden.

**Writing – original draft:** Maximiliane L. Verfuerden.

**Writing – review & editing:** Maximiliane L. Verfuerden, Sarah Dib, John Jerrim, Mary Fewtrell, Ruth E. Gilbert.

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
