## [Decision Letter · Decision Letter 0]

13 Aug 2020

PONE-D-20-14949

Effect of long-chain polyunsaturated fatty acids in infant formula on long-term ‎cognitive function in childhood: a systematic review and meta-analysis of randomised controlled trials

PLOS ONE

Dear Dr. Verfürden,

Thank you for submitting your manuscript to PLOS ONE. After careful consideration, we feel that it has merit but does not fully meet PLOS ONE’s publication criteria as it currently stands. Therefore, we invite you to submit a revised version of the manuscript that addresses the points raised during the review process.

ACADEMIC EDITOR

First, I must praise the authors for their extensive efforts in this work.

Second, some clarifications and improvements are still needed, as follows:

1- Are these figures produced by the same software mentioned in the methods, RevMan 5.3?

2- Why did you use the fixed effect model? I would advise using the random effect model instead owing to the present heterogeneity.

3- Publication bias assessment is not reliable in case of fewer than 10 included studies (according to egger et al). I advise you mention it only in the supplementary files (no need to be added to the full article).

4- I want to check the extracted data to repeat the analysis and compare with the reported results. Unfortunately, the raw data do not exist in the supplementary file. Please, provide them.

5-  When did you use MD and when did you use SMD? Why two formats of the Effect Estimate?

6- Did these studies report pre/post outcomes? Did you calculate the effect size based on (post - pre) or you consider the post values only for effect size calculation?

7- Cochrane Risk of Bias tool includes "blinding of study personnel", "blinding of outcome assessors" and "other bias" but I did not see these domains in the authors manuscript. You should include these domains in the ROB or clarify on-which basis they were omitted.

We look forward to receiving your revised manuscript.

Kind regards,

Ahmed Negida, MD

Academic Editor

PLOS ONE

Journal Requirements:

2. In your Methods section, please provide additional information about the search performed (we note that the full search string is reported as a supplementary table, but would suggest that more information is included in the main text).

3. Please ensure that every statement is supported by an appropriate reference. For example, we note that the statement "No systematic review has previously evaluated the effects of LCPUFA on cognitive function in infant formula beyond early childhood (age �2.5 years) when cognitive tests are more accurate" is not followed by any citation corroborating the information on cognitive tests.

5.Thank you for stating the following in the Financial Disclosure section:

[This study was supported by funds from the Economic and Social Research Council, and the Great Ormond Street Hospital Charity. RG was supported by Health Data Research UK, an initiative funded by the UK Research and Innovation, Department of Health and Social Care (England) and the devolved administrations, and leading medical research charities. Research at UCL Great Ormond Street Institute of Child Health is supported by the NIHR Great Ormond Street Hospital Biomedical Research Centre. ]. 

We note that one or more of the authors have an affiliation to the commercial funders of this research study : Health Data Research UK

Reviewers' comments:

Reviewer's Responses to Questions

**Comments to the Author**

1. Is the manuscript technically sound, and do the data support the conclusions?

Reviewer #1: Yes

Reviewer #2: Yes

Reviewer #3: Yes

Reviewer #4: Partly

2. Has the statistical analysis been performed appropriately and rigorously? 

Reviewer #1: Yes

Reviewer #2: No

Reviewer #3: N/A

Reviewer #4: No

3. Have the authors made all data underlying the findings in their manuscript fully available?

Reviewer #1: Yes

Reviewer #2: Yes

Reviewer #3: Yes

Reviewer #4: Yes

4. Is the manuscript presented in an intelligible fashion and written in standard English?

Reviewer #1: Yes

Reviewer #2: No

Reviewer #3: Yes

Reviewer #4: Yes

5. Review Comments to the Author

Reviewer #1: The authors performed a systematic review and meta-analysis of RCTs to assess the effect of LCUFA on long-term cognitive function in childhood. The authors concluded that “there was no evidence of benefit of LCPUFA supplementation in term or preterm-born infants and weak evidence that LCPUFA reduced IQ score in term-born children.”

Although the overall approach to the review is proper, I have few comments:

1. Abstract: Well-written structured abstract.

- I would recommend adding the software used for analysis.

- Please place the keywords after the abstract.

2. Introduction: The research question was clearly outlined, and the research question justified given what is already known about the topic.

- The text starts with references [3-5]. References should be numbered in consecutive order in the main text starting from “Introduction.”

3. Methods:

- The methodology described in the study seems to have followed current standardized procedures and guidelines for systematic review and meta-analysis. The methods are detailed enough to allow replication of the analysis. Outcomes are well-defined; the same is true for quality assessment and data analysis.

- Page 5: please define the PRISMA guidelines “Preferred Reporting Items for Systematic Reviews and Meta-Analyses.”

- Page 6: The paragraph starting with “The secondary outcomes…”, delete “standardised mean difference” because it is previously defined.

4. Results: The results are adequately presented.

- “We included eight unique trial cohorts,” please add references.

- a comma should be placed between the effect size and the 95% Cl.

5. Discussion: The results are discussed from multiple angles and placed into context without being overinterpreted, and conclusions answer the aims of the study.

6. Figure 1:

- The “Cochrane Central Register” search results are missing.

- 49 full-text articles assessed, the authors excluded 40 articles (27+13); thus, 9 articles should be retrieved, not 8. Please recheck!!

Reviewer #2: Abstract:

1- Please follow the guidelines of the journal (https://journals.plos.org/plosone/s/submission-guidelines)

Introduction:

2- References 1 and 2 are missing.

3- Please add reference for this information "Human breast milk contains DHA, AA, and their fatty acid precursors"

4- Define this abbreviation "IQ" at the first mention.

Methods:

5- Section of Methods need to be rearranged; you can develop a new sub-section for inclusion and exclusion criteria.

6- "Patient involvement" No need for this section or you can move it before the references.

7- Please move this section to be before the references "Role of the funding source"

Results

8- "We obtained previously unpublished outcome data for two RCTs: Firstly, a two-centre trial of term babies in England18 19 provided unpublished follow-up data on IQ assessments using the Wechsler Preschool and Primary Scales of Intelligence-Revised (WPPSI-R) at age 4.5 years and the Wechsler Abbreviated Scales of Intelligence (WASI) at age 16 years. Secondly, a two-centre trial in England of babies born preterm provided unpublished data from their IQ assessments using the WASI at age 16 years" Is their any duplication here? Please rephrase to avoid any confusion.

9- Please mention the Study ID (last name and year) in the table 1.

10- Remove any question mark in the table 1 and replace it with NA or NR.

11- In Figure 1: Please add additional arms for cochrane library and unpublished data

12- Did you used R program or Review Manager as these figures not belong to Review Manager?

13- Try to perform a sensitivity analysis to overcome this heterogeneity

Discussion:

14- The discussion is very week and need to be supported with some clinical aspects.

Reviewer #3: This is a systematic review and metaanalysis in which the authors investigated the long-term cognitive function effects of long-chain polyunsaturated fatty acids in infant formula in childhood at the age of 2.5 years or older. Although the study is well-written and structured, the overall quality of evidence for the included RCTs is low due to heterogeneity and quality of included studies. Moreover, many studies have investigated that topic extensively with similar results. I do not think that this paper will add more information to the medical literature related to that topic.

Reviewer #4: The authors conducted a meta-analysis to evaluate long-term effects of long-chain polyunsaturated fatty acids on IQ beyond 2.5 years of age. The findings are interesting and consistent with the previous literature. Some comments warrant attention:

* Abstract

- Why fixed effect model despite obvious heterogeneity?

- Add the number of randomized children in the results.

- weak evidence of lower IQ in the supplemented group: But the 95% CI is not consistent with this interpretation. It crosses the null value of zero for a quantitative MD.

- A single meta-analysis was registered on PROPSERO with two numbers!!

* What this study adds/What is already known: These sections are not suitable for PLOS One. It seems the manuscript was submitted using the author instructions of another journal (Similar is the abstract).

* Introduction

- There is an obvious problem with referencing: Numbers are not organized; many references are old back to 1992.

- Because human breast milk contains LCPUFA, could the authors account for breast-fed children in their analysis?

- At the start of the third paragraph, the authors should add "To our knowledge, ".

- The rationale for the meta-analysis is adequately stated.

* Methods

- "proceedings from major scientific meetings of child nutrition" can you specify?

- I see that the authors sub-grouped according to preterm/term delivery. But are there any data on children with low birth weight?

- "using validated measures" which included?

- The authors did not provide information on their data extraction process!

- "The primary outcome was the pooled difference, presented as mean difference (MD) and standardized mean difference (SMD), between intervention and control arms, based on the measure of cognitive function reported most frequently among the included studies" this part is not clear!

- So you selected fixed effect model based on previous meta-analyses?

- The secondary tests were cognitive test scores?? All IQ scoring tests, because there are various types of cognitive tests?

- The authors used random effects model only for secondary outcome bacause the studies used various tests. But also, the heterogeneity should have been expected for the primary outcome based on other different variables between studies.

- Add version and source of RevMan software! These forest plots do not appear to be extracted from RevMan!

- Also, specify that ou used the updated Cochrane ROB. II tool.

- The authors evaluated publication bias using funnel plots. They should mention that this method lacks reliability for less than 10 included studies.

* Results

- The authors should comment briefly on ROB assessment results of the included trials.

- What was the 2.5 year follow-up age cut-off based on?

- The authors analyzed based on total IQ scores. But these tests often have sections for different cognitive functions. Could the authors extract any data in this regard?

- Considering the I2 values in the forest plot, re-analysis under random effects model is needed!

- "The upper confidence interval included no difference (i.e. zero), but the average effect favoured a reduction in IQ in babies randomised to LCPUFA". So, I believe it should be interpreted as no difference as the CI is the imp value to determine significance here.

- For outcomes with substantial heterogeneity, what did the authors do to solve this heterogeneity?

* Discussion

- Well-written and comprehensive.

6. PLOS authors have the option to publish the peer review history of their article (what does this mean?). If published, this will include your full peer review and any attached files.

Reviewer #1: No

Reviewer #2: No

Reviewer #3: No

Reviewer #4: **Yes: **Abdelrahman Ibrahim Abushouk

---

## [Author Response · Author response to Decision Letter 0]

15 Sep 2020

See attached point-by-point response document for the reviewers

---

## [Decision Letter · Decision Letter 1]

21 Oct 2020

Effect of long-chain polyunsaturated fatty acids in infant formula on long-term ‎cognitive function in childhood: a systematic review and meta-analysis of randomised controlled trials

PONE-D-20-14949R1

Dear Dr. Verfürden,

We’re pleased to inform you that your manuscript has been judged scientifically suitable for publication and will be formally accepted for publication once it meets all outstanding technical requirements.

Kind regards,

Ahmed Negida, MD

Academic Editor

PLOS ONE

Additional Editor Comments (optional):

Reviewers' comments:

Reviewer's Responses to Questions

**Comments to the Author**

1. If the authors have adequately addressed your comments raised in a previous round of review and you feel that this manuscript is now acceptable for publication, you may indicate that here to bypass the “Comments to the Author” section, enter your conflict of interest statement in the “Confidential to Editor” section, and submit your "Accept" recommendation.

Reviewer #1: All comments have been addressed

Reviewer #2: All comments have been addressed

Reviewer #4: All comments have been addressed

2. Is the manuscript technically sound, and do the data support the conclusions?

Reviewer #1: Yes

Reviewer #2: Yes

Reviewer #4: Yes

3. Has the statistical analysis been performed appropriately and rigorously? 

Reviewer #1: Yes

Reviewer #2: Yes

Reviewer #4: Yes

4. Have the authors made all data underlying the findings in their manuscript fully available?

Reviewer #1: Yes

Reviewer #2: Yes

Reviewer #4: Yes

5. Is the manuscript presented in an intelligible fashion and written in standard English?

Reviewer #1: Yes

Reviewer #2: Yes

Reviewer #4: Yes

6. Review Comments to the Author

Reviewer #1: The authors have addressed all my comments/suggestions. I found their responses quite satisfactory and the revised version has been much improved.

Reviewer #2: I recommend accepting this manuscript, as the authors addressed all comments, and enhanced the manuscript significantly.

Reviewer #4: The authors have adequately addressed my concerns and I endorse the manuscript in the current for for publication.

7. PLOS authors have the option to publish the peer review history of their article (what does this mean?). If published, this will include your full peer review and any attached files.

Reviewer #1: No

Reviewer #2: **Yes: **Eshak I. Bahbah

Reviewer #4: **Yes: **Abdelrahman Ibrahim Abushouk

---

## [Editor Report · Acceptance letter]

26 Oct 2020

PONE-D-20-14949R1 

Effect of long-chain polyunsaturated fatty acids in infant formula on long-term ‎cognitive function in childhood: a systematic review and meta-analysis of randomised controlled trials 

Dear Dr. Verfuerden:

I'm pleased to inform you that your manuscript has been deemed suitable for publication in PLOS ONE. Congratulations! Your manuscript is now with our production department. 

Kind regards, 

on behalf of

Dr. Ahmed Negida 

Academic Editor

PLOS ONE